# Altered Functional Connectivity Density in Type 2 Diabetes Mellitus with and without Mild Cognitive Impairment

**DOI:** 10.3390/brainsci13010144

**Published:** 2023-01-13

**Authors:** Dongsheng Zhang, Shasha Liu, Yang Huang, Jie Gao, Weirui Liu, Wanting Liu, Kai Ai, Xiaoyan Lei, Xiaoling Zhang

**Affiliations:** 1Department of MRI, Shaanxi Provincial People’s Hospital, Xi’an 710068, China; 2Department of Clinical Science, Philips Healthcare, Xi’an 710000, China

**Keywords:** type 2 diabetes mellitus, functional connectivity density, resting-state functional magnetic resonance imaging, neuroimaging

## Abstract

Although disturbed functional connectivity is known to be a factor influencing cognitive impairment, the neuropathological mechanisms underlying the cognitive impairment caused by type 2 diabetes mellitus (T2DM) remain unclear. To characterize the neural mechanisms underlying T2DM-related brain damage, we explored the altered functional architecture patterns in different cognitive states in T2DM patients. Thirty-seven T2DM patients with normal cognitive function (DMCN), 40 T2DM patients with mild cognitive impairment (MCI) (DMCI), and 40 healthy controls underwent neuropsychological assessments and resting-state functional MRI examinations. Functional connectivity density (FCD) analysis was performed, and the relationship between abnormal FCD and clinical/cognitive variables was assessed. The regions showing abnormal FCD in T2DM patients were mainly located in the temporal lobe and cerebellum, but the abnormal functional architecture was more extensive in DMCI patients. Moreover, in comparison with the DMCN group, DMCI patients showed reduced long-range FCD in the left superior temporal gyrus (STG), which was correlated with the Rey auditory verbal learning test score in all T2DM patients. Thus, DMCI patients show functional architecture abnormalities in more brain regions involved in higher-level cognitive function (executive function and auditory memory function), and the left STG may be involved in the neuropathology of auditory memory in T2DM patients. These findings provide some new insights into understanding the neural mechanisms underlying T2DM-related cognitive impairment.

## 1. Introduction

Type 2 diabetes mellitus (T2DM) is a heterogeneous metabolic disorder characterized by reduced insulin sensitivity and relative insulin deficiency. The number of patients with T2DM worldwide is approximately 445 million, and the incidence of this disease is increasing [1]. In addition to causing impairments in various cognitive domains, such as visual space, attention, also memory and executive function [1,2], T2DM also increases the risk of mild cognitive impairment (MCI) and conversion from MCI to dementia [3]. Cognitive function impairment can severely affect patients’ self-management ability and reduce their quality of life, imposing a substantial economic burden to the society and the patients’ families [4]. Cognitive scales are often used to assess cognitive impairment in clinical practice; however, assessments based on these scales are relatively subjective and cannot easily reveal early changes in cognitive impairment. Thus, the identification of the potential imaging features of early cognitive impairment in T2DM patients, understanding the processes underlying the changes leading to cognitive impairment, and the administration of effective interventions may delay or avoid the occurrence and development of cognitive impairment in T2DM patients [5,6]. Since abnormal changes in brain structure and function, especially the functional disconnection of brain hub nodes, are the neural basis of cognitive impairment [7], the exploration of the change patterns in brain function in T2DM patients with different cognitive states can help reveal the neural mechanisms underlying cognitive impairment in T2DM patients. Although multiple teams have previously used resting-state electroencephalography (EEG) [8], transcranial magnetic stimulation (TMS) [9], positron emission tomography (PET) [10], and other methods to explore the neural mechanisms underlying cognitive impairment in T2DM, the vast majority of studies assessing structural and functional changes in the brain during different cognitive states in T2DM patients used magnetic resonance imaging (MRI) techniques. Multiple studies [11,12,13,14,15] have shown no significant differences or only mild changes in the white matter (WM) fiber tract integrity and WM network between T2DM patients with normal cognitive function (DMCN) and healthy controls (HCs), while T2DM patients with MCI (DMCI) showed extensive WM tract generation and the severe impairment of small-world network properties. Moreover, the WM integrity of the major interlobar pathways of the temporal lobe (right inferior frontal-occipital and inferior longitudinal fasciculus) has been shown to be significantly associated with impaired episodic memory and attentional function [11]. A voxel-based morphometry analysis [16] showed gray matter volume (GMV) atrophy in multiple brain regions, including the superior/middle temporal gyrus (STH/MTG), fusiform, and cingulate gyrus in T2DM patients but also observed that these changes were more extensive in DMCI patients. Structural studies have shown that the WM microstructure and GMV damage gradually worsen along with cognitive impairment in T2DM patients, indicating that the underlying anatomical changes may be closely related to the process of silent progression from normal cognition to MCI in T2DM patients. However, functional studies have suggested that the altered brain functions in T2DM patients under different cognitive states are more complex and diverse. Yang et al. [17] evaluated brain network connectivity and found more extensive impairments in intra-network and inter-network connectivity in DMCI patients relative to DMCN patients and HCs. A recent study [18] suggested that DMCN patients mainly show compensatory enhanced neuronal activity and increased nodal characteristics in multiple brain regions, while DMCI patients show a coexisting brain function construction pattern of compensation and impairment but mainly impairment. Furthermore, Zhang et al. [19] found that changes in the salience network functional connectivity and GMV were non-linear and complex in T2DM patients with cognitive impairment. Despite these findings, however, the pattern of whole-brain functional connectivity changes in T2DM patients under different cognitive states remains unclear. As a voxel-wise, data-driven method, functional connectivity density (FCD) mapping is widely used to examine the density distribution of whole-brain resting functional connectivity. The global functional connectivity density (gFCD) reflects the brain’s information communication capability to a large extent, and brain regions with high gFCD are thought to be hubs connecting different functional specialization systems [20]. In addition, brain regions with short-range and long-range FCD are often specialized for modular information processing and integrative information processing, respectively [21]. Several studies [22,23] have confirmed that the balance of short-range and long-range FCD disorders is closely related to cognitive impairment. The T2DM-induced impairments in glucose homeostasis may disrupt the established balance of short- and long-range FCD [24]. Since the brain shows high energy consumption, it is vulnerable to fluctuations in plasma glucose levels. Therefore, the exploration of the altered patterns of FCD in T2DM patients under different cognitive states may better reflect the functional abnormalities caused by long-term abnormal glucose homeostasis.

A previous study [25] showed that T2DM patients exhibit increased short-range FCD and decreased long-range FCD, which may indicate a trade-off between energy-cost and network efficiency at the expense of losing cognitive function. However, multiple studies have shown that the features of T2DM-associated cognitive dysfunction differ depending on the stage of diabetes [26,27] and that the cognitive impairment caused by hyperglycemia may involve a complex process [28]. Therefore, this study aimed to use FCD evaluations to explore the functional architecture of the abnormal hub nodes in T2DM patients in different cognitive states, which will provide some clues for the identification of neuroimaging markers for the early assessment of T2DM-related cognitive impairment. We speculated that the patterns of FCD alterations are not the same in T2DM patients with different cognitive states and that abnormalities in brain regions may be more extensive in DMCI patients. Furthermore, these abnormal FCD alterations may correlate with clinical/cognitive variables.

## 2. Materials and Methods

### 2.1. Participants

This study adopted a cross-sectional design. Eighty participants with T2DM (39 DMCN and 41 DMCI patients; age, 42–68 years) were recruited from the Department of Endocrinology of Shaanxi Provincial People’s Hospital, and 40 euglycemic individuals (44–65 years of age; fasting glucose level, <6.1 mmol/L; HbA1c < 6.0%; no family history of diabetes) who underwent health examinations at our hospital during the same period were also enrolled as HCs. All of the participants were right-handed and had received at least 9 years of education. The patients met the 2014 American Diabetes Association diagnostic criteria [29] and were receiving stable treatment. The exclusion criteria for both groups were as follows: (1) age-related WM change (ARWMC) scale score of >2; (2) contraindicated for MRI examination or unable to cooperate with MRI examination; (3) any neurological or psychiatric diseases, such as Parkinson disease or major depression; (4) organic lesions of the brain, such as tumors, hemorrhage, or vascular malformation, and surgical history of neurological trauma; (5) presence of other endocrine or systemic organic diseases; and (6) alcohol dependence and other psychotropic substance abuse. The inclusion criteria for the DMCI group were as follows: (1) complaints of memory decline that could affect the maintenance of normal daily activities; (2) Mini-Mental State Examination (MMSE) score of >24 and Montreal Cognitive Assessment (MoCA) score of <26; and (3) a lack of any other physical or mental disorders that could lead to abnormal cognition.

On the day of the scan, all patients were routinely prescribed medication in accordance with the clinical treatment protocol to control blood glucose levels. They arrived at the MRI department between 6:30 and 7:00 p.m. after dinner. Only one patient was scheduled to undergo examinations each day to ensure that everyone underwent MRI scans for the same period and with a relatively stable blood glucose level. The testing procedure and scanning time of HCs were the same as those of T2DM patients. During the scan, all participants kept their eyes closed and remained calm and reported no discomfort. The study was approved by the ethics committee of Shaanxi Provincial People’s Hospital. The study protocol was explained in detail to all participants, and all participants provided written informed consent prior to participation.

### 2.2. Clinical and Neuropsychological Data

We obtained the medical history and clinical data of the patients from the medical records and questionnaires, and the clinical data of HCs from the outpatient medical examination center. Blood pressure was averaged over three measurements taken on the same day. The participants maintained a fasting state for 8–12 h, and their elbow venous blood was collected for laboratory tests, such as the measurement of the glycated hemoglobin (HbA1c), fasting blood glucose (FBG), triglyceride (TG), total cholesterol (TC), and low-density lipoprotein cholesterol (LDL-C) concentrations. In addition, postprandial blood glucose (PBG) data for T2DM patients were collected in accordance with the standard procedures. All participants underwent the following neuropsychological examinations: MMSE, the preferred scale to exclude dementia, and MoCA, a rapid screening assessment for MCI, were both used to assess general cognitive function; information processing speed and attention were tested by the Trail-Making Test A (TMT-A); executive function and visuospatial skills were evaluated by the Clock-Drawing Test (CDT); memory function was estimated by using the Rey Auditory Verbal Learning Test (RAVLT) and analyzing the total immediate recall and delayed recall scores. All participants completed the MMSE, MoCA, TMT-A, and CDT assessments. The RAVLT was not administered to the HCs, and only a subset of the T2DM patients (24 DMCN and 36 DMCI) completed the RAVLT. The neuropsychological tests were conducted by clinical neuropsychologists with at least 5 years of experience.

### 2.3. Resting-State fMRI Data Acquisition

MRI data were obtained using a 3.0 T scanner (Ingenia, Philips Healthcare, Best, The Netherlands) with a 16-channel phased-array head coil. Before scanning, the participants were instructed to lie supine, secure their heads with a sponge pad, wear earplugs to reduce noise effects, close their eyes, and keep their heads still. First, routine T2WI and T2-FLAIR sequences were obtained to exclude individuals with excessive WM changes and organic brain lesions. Then, the participants were instructed to keep their eyes closed and to remain awake and quiet. Resting-state functional blood oxygen-level-dependent (BOLD) images were obtained using a gradient-echo planar sequence with the following parameters: repetition time (TR) = 2000 ms, echo time (TE) = 30 ms, flip angle (FA) = 90°, no. of volumes = 200, thickness = 4 mm (no gap), no. of slices = 34, field of view (FOV) = 230 mm × 230 mm, and matrix = 128 × 128. Sagittal 3-dimensional T1-weighted imaging (T1WI) was performed using a fast spoiled gradient-echo sequence with the following parameters: TR = 7.5 ms, TE = 3.5 ms, FA = 8°, slice thickness = 0.55 mm (no gap), 328 sagittal slices, FOV = 250 × 250 mm, and matrix = 256 × 256.

### 2.4. Resting-State fMRI Data Analysis

Functional data were preprocessed using DPARSF_V4.3 (http://www.restfmri.net/forum/DPARSF (accessed on 18 August 2021)) on the basis of MATLAB R2014b (MathWorks, Natick, MA, USA) with the following steps: (1) The first ten time frames were discarded to ensure that the signal reached equilibrium and showed saturation effects; (2) slice-timing correction was performed for interleaved acquisitions to correct the time delay between slices; (3) 3D head motion correction was performed, and patients with large head movement were excluded (head motion > 1.5 mm and/or translation > 1.5° of rotation in any direction); (4) the images were spatially normalized into a standard stereotaxic space at 3 × 3 × 3 mm, based on the Montreal Neurological Institute (MNI) EPI template; (5) the nuisance variables, including the cerebrospinal fluid and white matter signals, 24 head motion parameters, and the linear trend signals were regressed out from further analysis. Then, typical temporal band-pass filtering (0.01–0.08 Hz) was used to reduce the effect of very low-frequency drift and high-frequency physiological noise (Figure 1).

Subsequently, based on a previous study by Tomasi and Volkow [30], we used custom-written software in the Neuroscience Information Toolbox (NIT) to evaluate FCD mapping. Specifically, voxel-wise functional correlation analysis was conducted for each voxel using Pearson correlation analyses within the gray matter mask. Two voxels with a correlation coefficient of R > 0.6 were considered to be significantly connected [22]. The global FCD was computed by counting the number of functional connections between the given voxel and the whole-brain voxels. The short-range FCD calculated the correlation coefficient between a given voxel and its immediate neighbors. The voxel with an over-threshold connection for the given voxel was added to its neighboring cluster. Next, the same calculation was performed for each voxel in the neighboring cluster to expand the size of the neighboring cluster until no additional voxels were added. Then, the number of voxels in the final neighboring cluster were used to map the short-range FCD, which were defined on the basis of a neighborhood strategy and represents the intraregional connectivity. The long-range FCD was obtained by subtracting the short-range FCD from the global FCD, which represented interregional connectivity [31]. Finally, the resting-state FCD maps were spatially smoothed using a 6 × 6 × 6 mm full-width at half maximum (FWHM) Gaussian kernel and converted into *Z*-score maps for further statistical analyses.

### 2.5. Statistical Analysis

Analyses of demographic and clinical data among the three groups were performed with SPSS version 20.0 (SPSS Inc., Chicago, IL, USA). The Chi-squared (χ^2^) test was used to analyze sex-based differences; one-way analysis of variance (ANOVA) was performed with other data in the three groups; the least significant difference (LSD) was evaluated to perform post-hoc comparisons. PBG, RAVLT immediate and recall scores, and disease duration were assessed by independent two-sample *t*-tests in the two T2DM groups. *p* < 0.05 was considered statistically significant.

For the FCD maps, ANOVA and random-effect two-sample t-tests were performed in DPABI to depict the between-group differences in short-range FCD and long-range FCD with education level as covariates (GRF corrected *p* < 0.005, cluster level *p* < 0.05). Mean *Z*-scores for FCD were extracted from brain regions showing differences between groups to explore the relationships with clinical/cognitive scores after controlling for education (Bonferroni correction, *p* < 0.05).

## 3. Results

### 3.1. Comparison of Clinical and Neuropsychological Data

Three participants were excluded due to head motion (two patients with DMCN) and small-vessel disease (one patient with DMCI). Finally, a total of 77 patients with T2DM (37 DMCN patients and 40 DMCI patients) and 40 HCs were enrolled in the study (Figure 2). The demographic, clinical, and neuropsychological data of the patients with T2DM and the HCs are summarized in Table 1. All three groups showed no significant differences in sex, age, education, blood pressure, body mass index (BMI), CDT scores, or TG, TC, and LDL-C levels (*p >* 0.05), while the two groups of diabetic patients also showed no significant differences in the diabetes duration and PBG levels (*p >* 0.05). In comparison with HCs, the two T2DM groups showed higher FBG and HbA1c levels, and the DMCI group showed fewer years of education. The DMCI group also showed poorer MMSE and MoCA scores and higher TMT-A scores than the DMCN group and HCs and poorer RAVLT immediate and delay recall scores than the DMCN group.

### 3.2. FCD Analysis

The regions showing differences in the long-range FCD among the three groups were the left cerebellar lobule VIII/Crus II, left inferior temporal gyrus(ITG)/fusiform gyrus(FG), right inferior frontal gyrus (IFG), and left superior temporal gyrus (STG). In comparison with the HCs, the DMCN group showed lower long-range FCDs in the right MTG/ITG and left cerebellar lobule VIII/Crus I/II, while the DMCI group showed lower long-range FCDs in the left ITG, right IFG, and left cerebellar lobule VIII. The DMCI group also showed a lower long-range FCD in the left STG than the DMCN (Table 2, Figure 3). The regions showing differences in the short-range FCD among the three groups were the right MTG/ ITG, right MTG/ITG/FG /cerebellar lobule VI, right IFG, and left STG. In comparison with HCs, the DMCN group showed lower short-range FCDs in the left MTG/ITG and higher short-range FCDs in the bilateral middle cingulate gyrus (MCC)/right precuneus, while the DMCI group showed lower short-range FCDs in the right IFG, right MTG/ITG, and right ITG/FG/cerebellar lobule VI. The short-range FCDs showed no significant differences between the two T2DM groups (Table 3, Figure 4).

### 3.3. Correlation Analysis

After controlling for education, the long-range FCDs in the left STG and the RAVLT immediate (*r* = 0.356, *p* = 0.005) and delayed recall (*r* = 0.335, *p* = 0.009) scores showed significant positive correlations in all T2DM groups (Figure 5). Abnormally altered FCDs and other clinical/cognitive variables did not show significant correlations.

## 4. Discussion

The results of our study indicated that the patterns of functional architecture in T2DM patients with different cognitive statuses were somewhat similar but not completely consistent. Abnormal functional connectivity in T2DM patients mainly occurred in the temporal lobe and cerebellum, but the abnormal functional architecture in DMCI patients was more extensive. These findings were not consistent with those reported by Zhou et al. [32], who found that the global and local efficiency and multiple nodal centrality were significantly higher in DMCI patients relative to HC, whereas the whole-brain network topological properties were not significantly abnormal in DMCN patients. These differences between the studies may be attributable to the small number of participants or the inadequate sensitivity of global topological measurements to detecting minor changes in the early disease stage.

In this study, both long- and short-range FCD were bilaterally reduced in the MTG/ITG of patients with T2DM, indicating the impaired functional connectivity of these regions in both whole-brain functional integration and local modular information processing. Although the decreased short-range and long-range FCDs of the MTG/ITG in both T2DM groups were not on the ipsilateral cerebral hemisphere, studies [33,34] have confirmed that the bilateral hemispheric functions in these regions do not differ much. The MTG has been identified as a network hub of semantic processing and mainly contributes to controlled semantic retrieval processes [35,36]. The MTG shows increased neural activity during semantically demanding tasks [37,38]. Furthermore, the application of inhibitory repetitive TMS to the MTG has been shown to affect semantic control function, including performance on thematic and taxonomic tasks [39]. Although behavioral studies [40,41] have shown that T2DM patients have semantic cognitive deficits, the neural mechanisms underlying these deficits have not yet been elucidated. Our results may provide some clues for the further exploration of semantic cognitive dysfunction in T2DM patients. Several previous neuroimaging studies [18,42,43] have demonstrated functional impairments of the occipital primary visual cortex in T2DM patients. The regions of cerebellar lobule VI, FG, ITG are engaged in visual cognitive processes [44,45,46]. The ITG is an important part of the ventral visual pathway, and information from the primary visual cortex is transmitted to the ITG through the ventral visual pathway, culminating in high-level visual representations [47,48]. Patients with lesions in the ITG often exhibit deficits in object, face, color, or scene vision [49,50]. Our results for disordered ITG functional connectivity may indicate abnormal visual cognition in T2DM patients, consistent with the findings of previous studies by Xiong et al. [18].

Cerebellar lobule VIII and crus I/II belong to the posterior cerebellum, which are closely related to sensorimotor tasks, and several studies [51,52] have confirmed that the posterior cerebellum is susceptible to diabetes-related disruptions. Movement impairments in individuals with diabetes have historically been attributed to diabetic peripheral neuropathy (DPN). Our study included 46 T2DM patients with DPN, which may be the reason for the aberrant FCDs in the cerebellum. To confirm this hypothesis, patients with DPN were selected and compared with HCs, and the findings showed more extensive cerebellar sensorimotor region abnormalities in the T2DM patients with DPN (Appendix A). Furthermore, previous study has demonstrated abnormalities in cerebellar–cerebral circuits, including the motor pathways, and their associations with cognitive impairment [53]. However, motor impairments, including poor balance [54], altered gait [55], and compromised grip control [56], also occur in individuals with diabetes without DPN. Therefore, future studies should focus on the presence of the abnormal central regulation of sensorimotor function in patients without DPN, which may contribute to the understanding the neural mechanisms underlying diabetes-related sensorimotor impairment.

The precuneus and MCC are the two core regions of the parietal memory network (PMN) [57]. Numerous studies [58,59,60,61,62,63] have demonstrated that these regions exhibit robust retrieval success effects across diverse memory tasks. One study [64] reported increased neuronal activity in the precuneus and selective improvement in episodic memory after rMTS treatment in AD patients. Moreover, the findings obtained in AD patients with different cognitive states showed that the memory network and functional resilience ensues in posterior regions (precuneus and MCC) and the cerebellum and that preclinical AD patients without cognitive impairment showed increased compensatory functional coupling between the precuneus and MCC [65]. Therefore, we hypothesized that the increased short-range FCDs in the MCC and right precuneus in DMCN patients may be compensatory changes to memory impairment. A previous study [18] has also described the possibility of multiple compensatory mechanisms in altered brain function and reported increased precuneus node properties in DMCN patients.

Executive function is a common cognitive impairment domain in T2DM patients [66]. The right IFG is a central region for executive control [67] and is involved in a variety of higher cognitive functions such as working memory and attention [68,69,70]. A randomized controlled trial in patients with traumatic brain injury revealed that music-based rehabilitation enhances executive functioning in conjunction with the increase of GMV specifically in the right IFG [71]. Previous studies [72,73,74] have also suggested decreased GMV and aberrant neuronal activity in the IFG in T2DM patients, indicating that the IFG was a region susceptible to T2DM-related brain damage. One recent study [15] further confirmed executive dysfunction impairment correlated with nodal efficiency in the right IFG in DMCI patients. Therefore, we hypothesized that the reduced gFCD of the right IFG in this study may indicate abnormal executive function in DMCI patients. Moreover, the DMCI patients in the present study showed worse psychomotor speed and attention, which is consistent with the findings of previous studies [75,76] reporting that MCI patients often show impairment in multiple cognitive domains.

The left STG is a shared substrate for auditory short-term memory and speech comprehension, and the structural integrity of the STG and sulcus has been shown to predict auditory short-term memory capacity [77]. A recent meta-analysis [78] with a large sample size demonstrated that the STG is a robust brain region with reduced resting-state brain activity and may serve as a biomarker for the early diagnosis of MCI. Structural studies have shown that the average thickness of the left STG and bilateral entorhinal cortex plays a key role in the memory domain decline in MCI patients [79]. In functional studies, MCI patients showed the hyperactivation of the left STG and insula during executive functioning tasks of working memory, which may indicate compensatory responses to AD pathology [80,81]. The DMCI group showed decreased long-range FCDs in the left STG in comparison with the DMCN group, which may indicate the impaired cognitive function of auditory memory in DMCI patients. In addition, the z-scores of long-range FCDs in the left STG were positively correlated with the RAVLT immediate recall and long-term delayed recall scores, which further confirms our speculation.

Our study had several notable limitations. First, it was a cross-sectional study with a small sample size, which may have influenced the statistical power to some extent. Second, the treatment regimens of T2DM patients were not consistent, and although this inconsistency was unavoidable, it may have introduced some bias in the results. Third, the fields covered by our cognitive scale were not comprehensive enough, and future studies should aim to include more comprehensive and effective cognition-related scales to evaluate the visual and semantic functions of the participants. Fourth, the results of resting-state fMRI may lack specificity, and the use of task-state MRI for specific cognitive functions could validate our speculations.

## 5. Conclusions

To our knowledge, the present study is the first to explore the pattern of altered whole-brain FCD in T2DM patients with different cognitive states. The results showed that the FCD change characteristics in T2DM patients differed according to their cognitive state. The functional architecture disorders in DMCI patients were more extensive and included more abnormalities in brain regions corresponding to higher-level cognitive function (executive function and auditory memory function). In addition, the left STG may be involved in the neuropathology of auditory memory in T2DM patients, providing some new insights into the neural mechanisms underlying T2DM-related cognitive impairment.

## Figures and Tables

**Figure 1 brainsci-13-00144-f001:**
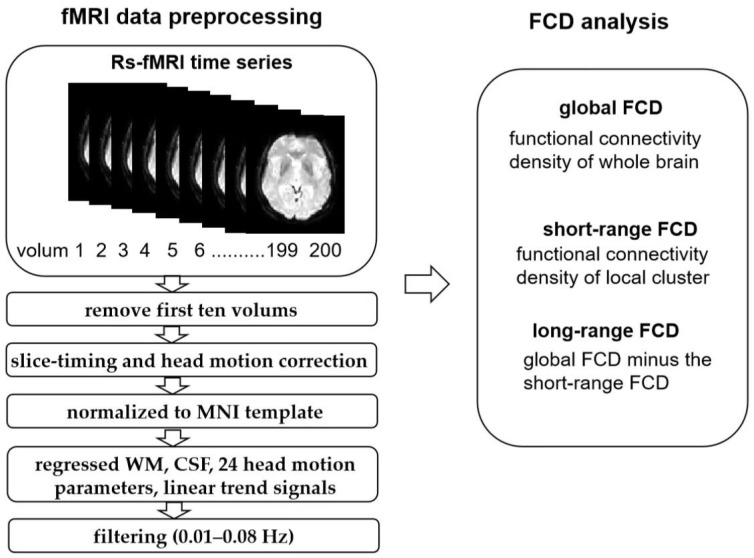
The fMRI data processing process in FCD.

**Figure 2 brainsci-13-00144-f002:**
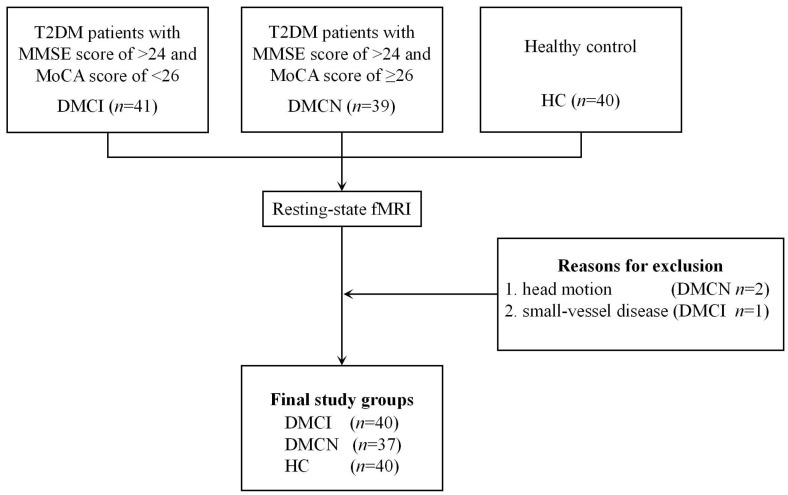
Flowchart of participant inclusion in the final registry and the exclusion criteria.

**Figure 3 brainsci-13-00144-f003:**
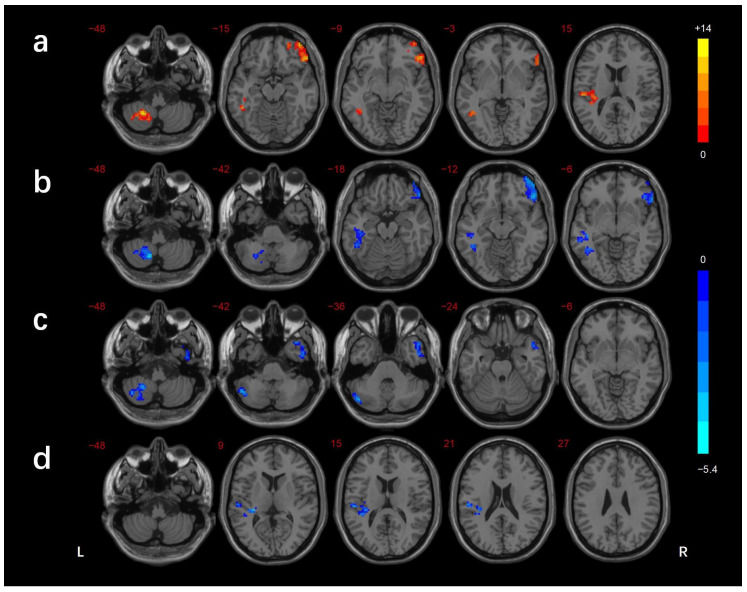
Comparison of long-range FCDs among groups (GRF corrected *p* < 0.005, cluster level *p* < 0.05). (**a**) Significant difference in long-range FCDs among three groups. (**b**) Significant difference in long-range FCDs between the DMCI and HC groups. (**c**) Significant difference in long-range FCDs between the DMCN and HC groups. (**d**) Significant difference in long-range FCDs between the DMCI and DMCN groups. The color scale denotes the *t*-value. L, left; R, right.

**Figure 4 brainsci-13-00144-f004:**
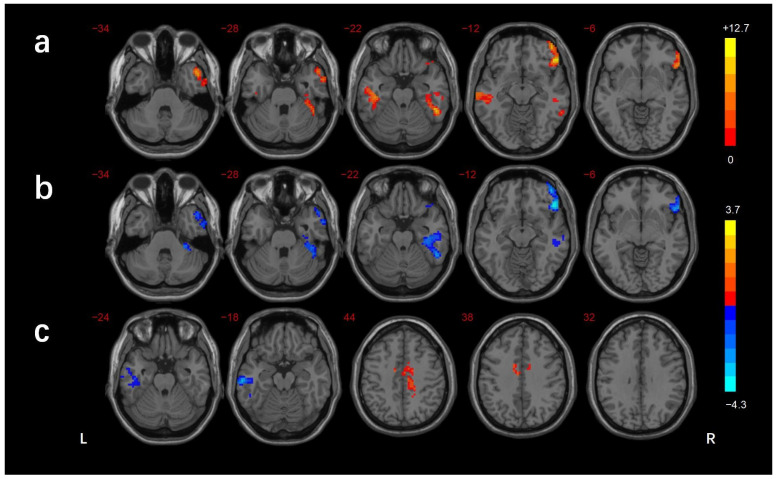
Comparison of short-range FCDs among groups (GRF corrected *p* < 0.005, cluster level *p* < 0.05). (**a**) Significant difference in short-range FCDs among three groups. (**b**) Significant difference in short-range FCDs between the DMCI and HC groups. (**c**) Significant difference in short-range FCDs between the DMCN and HC groups. The color scale denotes the *t*-value. L, left; R, right.

**Figure 5 brainsci-13-00144-f005:**
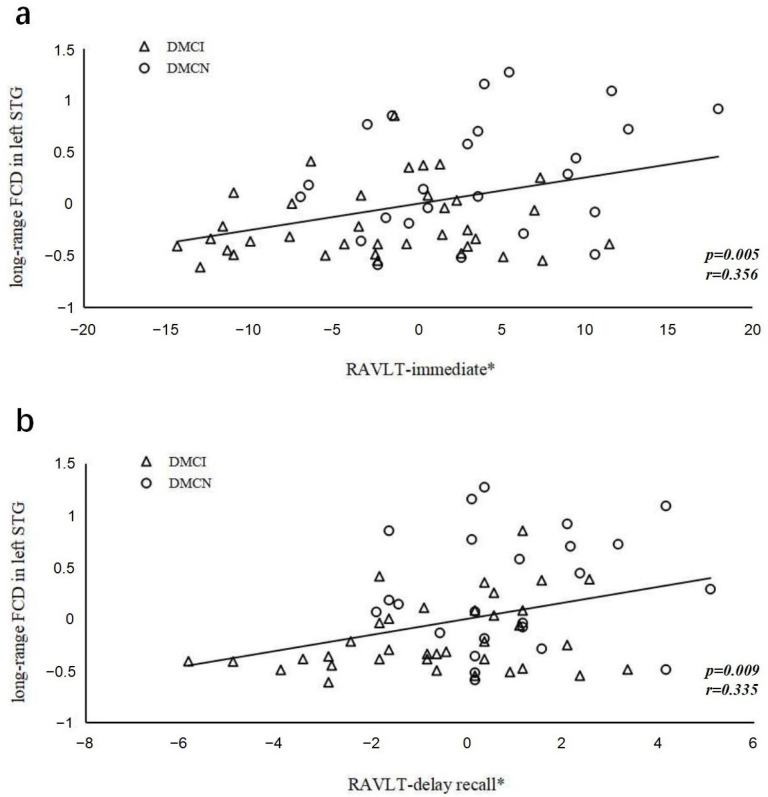
The correlations between RAVLT scores and long-range FCD in the left superior temporal gyrus. (**a**) Correlation between Z-scores of FCDs in the left superior temporal gyrus and the RAVLT immediate scores (*r* = 0.356, *p* = 0.005); (**b**) Correlation between Z-scores of FCDs in the left superior temporal gyrus and the RAVLT delayed recall scores (*r* = 0.335, *p* = 0.009). The asterisk (*) indicates coordinate values, controlling for effects of years of education.

**Table 1 brainsci-13-00144-t001:** Demographic, clinical, and neuropsychological data of T2DM patients and healthy controls.

	HC (*n* = 40)	DMCN (*n* = 37)	DMCI (*n* = 40)	*F/χ^2^*	*p*
Age (years)	54.80 ± 5.35	54.03 ± 5.86	55.33 ± 6.60	0.460	0.633
Sex (male/female)	27/13	26/11	24/16	0.978	0.613 ^#^
Duration (years)	-	9.05 ± 5.60	9.48 ± 5.44	-	0.739 ^&^
Education (years)	14.35 ± 2.82	14.08 ± 2.20	13.18 ± 2.71 ^a^	2.229	0.112
BMI (kg/m^2^)	24.41 ± 2.93	25.05 ± 2.67	24.75 ± 2.77	0.456	0.635
Systolic blood pressure (mmHg)	124.62 ± 12.27	126.14 ± 18.77	127.28 ± 16.82	0.175	0.840
Diastolic blood pressure (mmHg)	85.47 ± 9.40	81.38 ± 9.92	80.68 ± 13.79	0.944	0.393
FBG (mmol/L)	5.19 ± 0.81	8.78 ± 2.85 ^a^	8.21 ± 2.825 ^a^	19.977	<0.001 *
PBG (mmol/L)^2^	-	12.76 ± 3.20	12.14 ± 3.78	-	0.498 ^&^
HbA1c (%)	5.62 ± 0.51	8.26 ± 1.66 ^a^	7.90 ± 2.37 ^a^	25.336	<0.001 *
TG (mmol/L)	1.83 ± 1.25	2.55 ± 4.46	1.86 ± 0.86	0.858	0.427
TC (mmol/L)	4.87 ± 0.95	4.65 ± 1.56	4.52 ± 1.20	0.741	0.479
LDL-C (mmol/L)	2.74 ± 0.77	2.57 ± 0.66	2.42 ± 0.88	1.491	0.230
MMSE score	28.69 ± 1.49	28.68 ± 1.73	27.65 ± 1.42 ^a,b^	5.834	0.004 *
MoCA score	26.90 ± 1.92	27.43 ± 1.06	22.95 ± 1.92 ^a,b^	81.722	<0.001*
CDT score	22.56 ± 6.58	22.50 ± 7.66	21.46 ± 8.12	0.260	0.772
TMT-A score	68.11 ± 26.93	74.68 ± 31.92	88.10 ± 28.85 ^a,b^	4.745	0.011 *
RAVLT immediate score	-	46.29 ± 6.86	40.03 ± 6.59	-	0.001 ^&^
RAVLT delay score	-	9.71 ± 1.94	8.11 ± 2.10	-	0.004 ^&^

Data are presented as mean ± standard deviation or number (%) unless otherwise indicated. BMI, body mass index; FBG, fasting blood glucose; PBG, postprandial blood glucose; HbA1c, glycated hemoglobin; TG, triglyceride; TC, total cholesterol; LDL-C, low-density lipoprotein cholesterol; MMSE, Mini-Mental State Examination; MoCA, Montreal Cognitive Assessment; CDT, Clock-Drawing Test; TMT-A, Trail-Making Test A; RAVLT, Rey Auditory Verbal Learning Test. ^#^ The *p*-value was obtained using the χ2 test. ^&^ The *p*-value was obtained using the independent two-sample *t*-test. ^a^ post hoc paired comparisons show significant differences compared with HCs; ^b^ post hoc paired comparisons show significant differences compared with DMCN patients; * *p* < 0.05.

**Table 2 brainsci-13-00144-t002:** Regions of abnormal long-range FCD among the three groups.

Brain Regions	BA	Voxels (mm^3^)	Peak MNI Coordinates			f/*t*-Value
			X	Y	Z	
**ANOVA**						
L_cerebellar lobule VIII/Crus II	-	104	−24	−54	−48	13.916
L_ITG/FG	37	109	−39	−51	−6	12.070
R_IFG	47	217	51	33	−12	13.389
L_STG	13/41	116	−36	−30	9	12.689
**DMCN vs. HC**						
L_cerebellar lobule VIII/Crus I/II	-	159	−48	−72	−33	−4.836
R_ MTG/ITG	38/21	175	51	12	−30	−4.089
**DMCI vs. HC**						
L_cerebellar lobule VIII	-	125	−15	−63	−48	−5.304
L_MTG/ITG/FG	37	245	−45	−60	0	−4.382
R_IFG	47	253	51	33	−15	−4.695
**DMCI vs. DMCN**						
L_STG	41	124	−30	−30	9	−3.784

MTG, middle temporal gyrus; ITG, inferior temporal gyrus; STG, superior temporal gyrus; IFG, inferior frontal gyrus; MCC, mid-cingulate cortex; FG, fusiform gyru; BA, Brodmann’s area; MNI, Montreal Neurological Institute; L, left; R, right. Group differences in functional connectivity were evaluated by two-sample *t*-tests (*p* < 0.05, Gaussian random field-corrected).

**Table 3 brainsci-13-00144-t003:** Regions of abnormal short-range FCDs among the three groups.

Brain Regions	BA	Voxels (mm^3^)	Peak MNI Coordinates			f/*t*-Value
			X	Y	Z	
**ANOVA**						
R_MTG/ITG	21/38	194	51	9	−39	10.514
R_MTG/ITG/FG/cerebellar lobule VI	37/20	187	54	−48	−18	11.278
L_ITG	21	155	−60	−21	−15	10.434
R_IFG	47	206	51	33	−15	12.651
**DMCN vs. HC**						
L_MTG/ITG	21	188	−66	−24	−15	−3.975
B_MCC/R_precuneus	24	188	15	−24	42	3.665
**DMCI vs. HC**						
R_MTG/ITG	21/38	185	51	12	−45	−4.018
R_MTG/FG/cerebellar lobule VI	37/20	299	39	−24	−18	−3.908
R_IFG	47	229	48	30	−12	−4.243

MTG, middle temporal gyrus; ITG, inferior temporal gyrus; IFG, inferior frontal gyrus; MCC, mid-cingulate cortex; FG, fusiform gyru; BA, Brodmann’s area; MNI, Montreal Neurological Institute; L, left; R, right. Group differences in functional connectivity were evaluated by two-sample *t*-tests (*p* < 0.05, Gaussian random field-corrected).

## Data Availability

The datasets used and analyzed during this study are available from the corresponding author on reasonable request.

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
