# Peer review of "Altered Functional Connectivity Density in Type 2 Diabetes Mellitus with and without Mild Cognitive Impairment"

_brainsci, 2023, doi:10.3390/brainsci13010144_

Round 1

Reviewer 1 Report

This study aimed to investigate Altered functional connectivity density in type 2 diabetes mellitus with and without mild cognitive impairment. I have the following suggestions.

1.    What is the novelty of this study, although several studies of functional connectivity in diabetes mellitus with and without mild cognitive impairment have been investigated earlier?

2.    Please write down the contribution of the study at the end part of the Introduction section in bulleted form.

3.    The abstract should be improved by combining the objectives, short methodology, main findings result, and prospective application.

4.    Authors should include a conceptual figure of their proposed approach with more details and parametrization.

5.    Manuscript is difficult to follow and need improvement in writing and drawing figures.

6.    Authors should add a figure of the experimental protocol in this study.

7.    Authors should describe more details of the MRI data pre-processing, MRI feature extraction and data analysis.

8.    Authors should introduce the mental workload, cognitive impairment using other methods, such as, 2-D brain imaging EEG. Machine-learning approaches are utilized for stroke prediction in article, healthsos: real-time health monitoring system for stroke prognostics; in article, quantitative evaluation of task-induced neurological outcome after stroke; in article, quantifying physiological biomarkers of a microwave brain stimulation device; in article, quantitative evaluation of eeg-biomarkers for prediction of sleep stages; and in article, driving-induced neurological biomarkers in an advanced driver-assistance system.

9.    Quantification of dynamic brain activation pattern during a cognitive task is quite complex and likely influenced by many pathophysiological factors. Due to the very limited data, the results provided in this paper might not support the hypothesis sufficiently.

10.  The results and discussion section need to be extended and improved. Authors must make discussion on the advantages and drawbacks of their proposed method with other studies adding a discussion section.

11.  From the writing point of view, the manuscript must be checked for typos and the grammatical issues should be improved.

Author Response

Dear Madam or Sir, Thank you very much for giving us an opportunity to improve our manuscript entitled “Altered functional connectivity density in type 2 diabetes mellitus with and without mild cognitive impairment” (ID: brainsci-2050183). Please see the attachment for detailed modification.

Reviewer 2 Report

In this work, the authors described a study to characterize the neural mechanisms underlying type 2 diabetes mellitus (T2DM)-related brain damage and explored the altered functional architecture patterns in different cognitive states in T2DM patients. The method description was largely clear, but in my opinion the clinical motivation was not adequately focused, the paper appeared rushed and should be revised. I will try to explain my concerns in more detail below.

The introduction should provide the motivation for the work. Please include a more thorough description of the challenge you are trying to address and the potential impact of your work. More specifically, it would be useful to include statements describing the prevalence as well as the current clinical practice for diagnosing these patients, limitations of current methods, and the potential impact of your proposed method.

The discussion should reflect on findings in the current work. Please try to compare your results with others. A continuation of the introduction with further literature review should be done. Please reorganize these sections to make your paper even better.

Author Response

(The authors gave the same response as above.)

Round 2

Reviewer 1 Report

I have final comments below:

Spatial maps of the FCD of patients of the three groups should be reported.

Author Response

Dear Madam or Sir, Thank you very much for giving us an opportunity to improve our manuscript entitled “Altered functional connectivity density in type 2 diabetes mellitus with and without mild cognitive impairment” (ID: brainsci-2050183). We appreciate your constructive comments and suggestions on our manuscript. In this paper, the FCD analysis section of the results reports the long-range and short-range FCD result maps with differences among the three groups. In Figure 3,4, “ANOVA” is used to represent spatial maps of the FCD of patients of the three groups, so we did not modify the figures yet. Of course, if you think there is anything in the picture that needs to be modified, we will modify the picture according to your requirements.